# Cancer Stem Cell Functions in Hepatocellular Carcinoma and Comprehensive Therapeutic Strategies

**DOI:** 10.3390/cells9061331

**Published:** 2020-05-26

**Authors:** Yu-Chin Liu, Chau-Ting Yeh, Kwang-Huei Lin

**Affiliations:** 1Department of Biochemistry, College of Medicine, Chang-Gung University, Taoyuan 333, Taiwan; k1506820@gmail.com; 2Department of Biomedical Sciences, College of Medicine, Chang-Gung University, Taoyuan 333, Taiwan; 3Liver Research Center, Chang Gung Memorial Hospital, Taoyuan 333, Taiwan; chauting@adm.cgmh.org.tw; 4Research Center for Chinese Herbal Medicine, College of Human Ecology, Chang Gung University of Science and Technology, Taoyuan 333, Taiwan

**Keywords:** hepatocellular carcinoma, liver cancer stem cells, stemness, self-renewal, tumorigenicity, therapeutic resistance

## Abstract

Hepatocellular carcinoma (HCC) is a significant cause of cancer-related mortality owing to resistance to traditional treatments and tumor recurrence after therapy, which leads to poor therapeutic outcomes. Cancer stem cells (CSC) are a small subset of tumor cells with the capability to influence self-renewal, differentiation, and tumorigenesis. A number of surface markers for liver cancer stem cell (LCSC) subpopulations (EpCAM, CD133, CD44, CD13, CD90, OV-6, CD47, and side populations) in HCC have been identified. LCSCs play critical roles in regulating HCC stemness, self-renewal, tumorigenicity, metastasis, recurrence, and therapeutic resistance via genetic mutations, epigenetic disruption, signaling pathway dysregulation, or alterations microenvironment. Accumulating studies have shown that biomarkers for LCSCs contribute to diagnosis and prognosis prediction of HCC, supporting their utility in clinical management and development of therapeutic strategies. Preclinical and clinical analyses of therapeutic approaches for HCC using small molecule inhibitors, oncolytic measles viruses, and anti-surface marker antibodies have demonstrated selective, efficient, and safe targeting of LCSC populations. The current review focuses on recent reports on the influence of LCSCs on HCC stemness, tumorigenesis, and multiple drug resistance (MDR), along with LCSC-targeted therapeutic strategies for HCC.

## 1. Introduction

Embryogenesis of both normal and tumor cells involves similar processes, including proliferation, motility, homing, dynamic morphologic changes, cellular heterogeneity, and interactions with the microenvironment. However, carcinogenesis is described as deregulation of malignant organogenesis regulated by abnormally proliferating and metastatic cancer and activated stromal cells that trigger angiogenesis, fibrosis, and inflammation [1]. One such case is liver cancer, which is classified as primary or secondary. Primary liver cancer refers to initiation of liver cell growth, and secondary liver cancer refers to spread of cancer cells to other organs from the liver. Primary liver cancer can be classified as growth of a single lump or growth in many places in the liver at the same time. Primary liver cancer types include hepatocellular carcinoma, cholangiocarcinoma, liver angiosarcoma, and hepatoblastoma. Hepatocellular carcinoma (HCC), also known as hepatoma, is the most common type worldwide, accounting for ~75% of all liver cancers. HCC is influenced by several important risk factors, with two distinct mechanisms of molecular pathogenesis: hepatitis infection (HBV or HCV) or toxin/environmental (alcohol or aflatoxin B) or metabolic (insulin resistance, obesity, type II diabetes or dyslipidemia in nonalcoholic HCC) factors that trigger liver tissue damage, leading to cirrhosis associated with hepatic regeneration and subsequent HCC [2] and genetic/epigenetic changes that influence the expression patterns of oncogenes or tumor suppressor genes [3,4,5,6,7]. The above factors are correlated with multiple dysregulated signaling pathways, such as growth factor-mediated angiogenic signaling (vascular endothelial growth factor (VEGF), platelet-derived growth factor (PDGF), epidermal growth factor (EGF), insulin-like growth factor (IGF), hepatocyte growth factor (HGF)/c-MET), mitogen-activated protein kinase (MAPK), phosphatidylinositol-3 kinase (PI3K)/AKT/mammalian target of rapamycin (mTOR), and Wnt/β-catenin pathways, which contribute to HCC development and tumorigenesis [8]. Elucidation of these signaling mechanisms is interesting from a therapeutic perspective, since targeting them may aid in reversing, delaying, or preventing the occurrence of HCC. Sorafenib is a first-line treatment approved by the United States Food and Drug Administration (USFDA) shown to benefit post-therapy survival rates in unresectable HCC cases. Subsequently identified target drugs, including regorafenib and lenvatinib, are currently used as second-line treatments for HCC. The above drugs can be effectively combined with radiation therapy and chemotherapy for clinical treatment of HCC. However, the therapeutic effects remain limited, which is ascribed to high recurrence and drug resistance of liver cancer stem cells (LCSCs), a subpopulation of liver cancer cells isolated via flow cytometry with self-renewal, differentiation, and tumorigenesis capabilities [9] hat play critical roles in tumor progression and therapeutic resistance. In this review, the functions of LCSCs in HCC and targeted therapeutic strategies are comprehensively discussed.

## 2. Identification and Plasticity of LCSCs

### 2.1. Concept of Cancer Stem Cells (CSCs)

Cancer stem cells (CSCs) have similar characteristics to normal stem cells, including self-renewal and differentiation. CSCs are also called as tumor-initiating cells (T-ICs) or cancer stem-like cells, which were first evidenced by injecting the AML cells into SCID mice by xenotransplant; the experiments indicated that expression of specific CSCs marker (CD34^+^CD38^−^) could promote production of large numbers of colony-forming progenitors [10]. This discovery suggested a new CSCs concept, according to which heterogeneity and tumor hierarchy is organized by a subset of cells with CSCs. This avoids traditional thoughts that heterogeneity is the progressive accumulation of multiple genetic [11] or epigenetic changes [12]. Several CSCs have been isolated from malignancies including lung cancer, pancreatic cancer, breast cancer, prostate cancer, colon cancer, glioma, and liver carcinoma [13,14,15,16]. CSCs have been found to possess highly tumorigenic, metastatic, and chemotherapy- and radiation-resistant properties, possibly leading to tumor relapse after therapy. CSCs evade multiple drug actions (MDR) with the aid of various intrinsic and external mechanisms [17]. Intrinsic mechanisms of chemoresistance include DNA damage repair pathway activation, high-level expression of drug efflux-related proteins, the capability of reconstituting original tumors, and the influence of epithelial-to-mesenchymal transition (EMT) and self-renewal-related genes [18]. External mechanisms of chemoresistance include activation of signaling pathways involved in epithelial-mesenchymal transition (EMT), hypoxia stimulation, and abnormal angiogenesis [17]. Besides, CSCs enter a dormant state (arrest in the G0 phase) of reduction of cell proliferation activity and persist resistance to chemotherapy for a few years, eventually leading to relapse. The relationship between tumor microenvironment (TME) and CSCs play an important role in influencing resistance therapeutics and promoting trans-differentiation of non-CSCs into CSCs by providing anti-apoptosis, stemness-maintaining factors, and matrix components. Thus, to study how to improve therapeutic resistance of CSCs in cancer is a critical issue.

### 2.2. Correlation between Hepatocellular Carcinoma and Cancer Stem Cells

CSCs of hepatocellular carcinoma (HCC) are termed liver cancer stem cells (LCSCs). LCSCs display specific features, such as the ability to generate new tumors displaying the phenotypes of xenotransplanted tumors, chemoresistance, metastasis, and recurrence [19] (Figure 1 and Table 1). Both metastasis and recurrence are associated with drug resistance [20]. So far, several surface markers and side population cells (SP) of HCC have been isolated, including EpCAM, CD133, CD44, CD13, CD90, CD24, CD47, and OV6. Other surface markers, such as K19, c-kit, ABCG2, and ALDH, have additionally been identified, which individually affect resistance to radiotherapy or chemotherapy and tumorigenesis by influencing drug efflux-related gene expression [21,22,23,24], activation of growth signaling, and stem cell-related and anti-apoptosis pathways [25,26,27]. LCSCs possess the capability of circulation within the body that significantly promotes distant metastasis and homing ability, compared to other tumor cell types. In other words, LCSCs promote tumor growth of primary cancer cells and metastasis of transplanted secondary tumors, leading to recurrence of HCC [28]. LCSCs are therefore closely related to metastasis, recurrence, and MDR and serve as an important diagnostic marker for HCC. In recent years, accumulating research has focused on identifying different surface markers of HCC through fluorescence-activated and magnetically activated cell sorting approaches, which could be effectively developed for eliminating LCSCs to achieve inhibition of tumor recurrence [29].

### 2.3. EpCAM 

The epithelial cell adhesion molecule, EpCAM, belonging to the type I transmembrane protein family is glycosylated and expressed in various tissues, including human epithelial and tumor tissues as well as progenitor/stem cells [63]. The EpCAM structure comprises an extracellular domain with epidermal growth factor (EGF) and thyroglobulin repeat-like domains, a single transmembrane domain, and a short 26-amino acid intracellular domain designated EpICD [63]. Interestingly, EpCAM is not only detectable in normal adult hepatocytes also expressed in embryonic liver, bile duct epithelium, and proliferating bile ductulus in cirrhotic liver and is thus considered a progenitor/stem cell marker in adult liver [64]. Data from systematic analyses suggest that EpCAM expression is essential for all human adenocarcinomas, including specific types of squamous cell carcinoma, retinoblastoma, and hepatocellular carcinoma [65,66]. EpCAM is not only involved in cell–cell adhesion, as the name indicates, but also cell proliferation, migration, cell cycle metabolism, signaling, differentiation, metastasis, regeneration, organogenesis, and tumorigenesis of the liver. EpCAM has additionally been detected on the surface of LCSCs and pancreatic CSCs [32,67]. Transplantation of isolated EpCAM^+^CD45^−^ cells from HCC patients into NOD/SCID mice initiated tumor formation, whereas EpCAM^−^CD45^−^ cells failed to form tumors, suggesting that EpCAM^+^ confers the stem/progenitor cell trait of HCC and promotes tumor growth [68]. EpCAM is involved in two major signaling pathways, specifically, intramembrane proteolysis and shedding of the extracellular domain [69]. EpICD is found in the cytoplasm generated from EpCAM cleavage by two important proteins (tumor necrosis factor-alpha converting enzyme (ACE) and presenilin 2 (PS-2)) and subsequently interacts with β-catenin through four-and-a-half LIM domain protein 2 (FHL2) [63]. Simultaneously, accumulation of β-catenin in the cytoplasm is dependent on inhibition of phosphorylation through induction of the β-catenin degradation complex (AXIN, APC, GSK3) by the Wnt signaling pathway [31]. Accumulated β-catenin interacts with FHL2 and EpICD to form a large protein complex that is translocated to the nucleus. The nuclear protein complex regulates transcription of EpCAM target genes, including c-myc, cyclins, and TCF1 [30,63]. Gene expression and pathway analyses suggest that activation of Wnt/β-catenin signaling enriches the EpCAM^+^ cell population. EpCAM^+^-rich cell subpopulations isolated from HCC present liver cancer stem cell features, which promote self-renewal, differentiation, and invasiveness [70]. Dt81 Hepa1-6, a new cell line derived from Hepa1-6 through in vivo passage in C57BL/6 mice, displays higher tumorigenicity, which is attributable to increased EpCAM and β-catenin expression [71]. In another study, EpCAM-positive circulating tumor cells were identified from HCC patients undergoing curative resection, which displayed stem cell-like and EMT phenotypes that were likely to cause tumor recurrence after surgical resection [68]. Additionally, stemness genes, such as Nanog, Sox2, and Oct4, were expressed in EpCAM-positive HCC cells and TSC2-AKT signaling activated upon Sorafenib treatment, further exacerbating hepatocellular carcinoma progression [33]. In another study, gene and protein expression profiles were analyzed from 245 and 144 hepatocellular carcinoma patients, respectively. EpCAM^+^ cells abundantly expressed CDH4, a chromatin remodeling enzyme, and influenced PPAR and DNA double-stranded break repairs to enhance chemoresistance of HCC cells [72]. Clinical analyses showed higher EpCAM expression in HCC tumor than adjacent normal liver tissue and positive correlation with differentiation grade among the clinicopathological parameters examined. Kaplan–Meier analyses showed that at advanced clinical stages, high EpCAM expression and poor differentiation grade were associated with poor survival rates. These results collectively support the utility of EpCAM in HCC as a predictive biomarker for unfavorable prognosis [73].

### 2.4. CD133

CD133 (human prominin-1, PROM1) is a glycoprotein with five transmembrane domains and two larger extracellular glycosylation chains that serves as an important surface marker, showing abundant expression in both cytoplasm and nucleus of various tumor tissues, including hepatocellular carcinoma (HCC), brain tumor, pancreatic cancer, prostate cancer, and colon cancer [74,75,76,77,78]. The CD133^+^ subpopulation was initially isolated from Huh7, one of the benign HCC cell lines, and was shown to play potential roles in proliferation and tumorigenesis [34] in both SMMC7721 and PLC8024 cells [35,79]. Additionally, 0~65% human HCC cells are CD133^+^. Knockout of CD133 in HCC cells is reported to reduce tumorigenicity and cell cycle progression [35]. Similar findings were obtained with clinical HCC patients, whereby high CD133 expression was correlated with poor prognosis. Simultaneously, increased CD133 levels in HCC patients were associated with shorter overall survival and higher recurrence rates, indicating that CD133 may be a suitable prognostic marker [80]. Recent studies suggest that CD133 induces differentiation into non-hepatocyte-like lineages and may act as a progenitor cell marker not only in damaged liver and HCC tissue but also cholangiocarcinoma, both in vitro and in vivo [81,82]. CD133 is involved in numerous molecular mechanisms, including self-renewal, multi-lineage differentiation, and tumorigenic and therapeutic resistance. Following EMT promotion of HBx-infected hepatoma cells, TGF-β expression of neighboring endothelial cells is increased, leading to significantly enhanced CD133 expression [36]. Aquaporin 3 (AQP3) is reported to maintain stemness and self-renewal capacity through inducing CD133 transcription activity via promoting stimulation and nuclear translocation of signal transducer and activator of transcription 3 (STAT3) and CD133 promoter-acetylated histone H3 [83]. In patients with hepatocellular carcinoma (HCC) recurrence, the majority of hepatic stem/progenitor cell (HSC/HPC) biomarkers are overexpressed, including cytokeratin 19, ABCG2, CD133, Nestin, and CD44, and angiogenesis agents CD34, VEGF, and PD-ECGF. Additionally, RFA-induced VEGF promotes tumor stemness and tumorigenesis via Nanog and activation of VEGFR2, which are positively correlated with CD133 expression in HCC tissues derived from patients with early recurrence. These results clearly suggest that VEGF and CD133^+^ stemness markers are positively correlated with early HCC recurrence [84]. Isolated CD133^+^ subpopulations of HCC cells and xenograft mouse models confer increased chemoresistance to doxorubicin (DOX) and fluorouracil (5-FU) through activation of Akt/PKB and Bcl-2 survival pathways [27]. Additionally, the epigenome cooperates with other regulatory factors, such as non-coding RNAs, allowing differentiation between short non-coding RNAs (miRNAs and sRNAs; <200 nucleotides) and long non-coding RNAs (lncRNAs; >200 nucleotides) [85]. MiR-130b overexpressed in CD133^+^ tumor-initiating cells (TIC) of HCC enhances chemoresistance, tumorigenicity, and self-renewal via suppression of its downstream target gene, TP53INP1 [86]. Dysregulation of lncRNAs in primary HCCs has additionally been shown to influence HCC tumorigenesis. 

### 2.5. CD44

CD44 is a transmembrane glycoprotein encoded by a single gene expressed as several isoforms that acts as a receptor for hyaluronic acid (HA). CD44 has been identified in both normal and tumor cells. Tumor cell behaviors, including proliferation, survival, migration/invasion, and chemoresistance, are closely associated with upregulation of CD44 [37,38,39,40]. Tumor-associated macrophages (TAM) are a class of immune cells often found in the microenvironment of solid tumors. TAMs produce IL-6 with subsequent activation of signal transducer and activator of transcription 3 (STAT3) and promote expansion of CD44^+^ populations and tumorigenesis in cultures as well as growth of xenograft tumors in mice [87]. CD44s, one of the isoforms of CD44, plays a completely opposite role to CD44. Knockout of the CD44 gene in HCC expressing CD44s only resulted in decreased maintenance of CSCs and increased drug sensitivity [88]. CD44 overexpression in HCC patients and cell lines is regulated by TGF-β and confers a TGF-β–mediated mesenchymal phenotype through increased AKT/GSK-3β/β-catenin activity [42], with a further major influence on EMT-related genes, such as positive correlation with vimentin expression, negative correlation with E-cadherin expression, high percentage of phospho-Smad2–positive nuclei, and reduced disease-free and overall survival [89]. Simultaneous analysis of clinicopathological parameters of other patient groups revealed that CD44 is also involved in local aggressive HCC recurrence through regulation of EMT genes (including low E-cadherin, high vimentin, and high N-cadherin expression) after local ablation therapy (LAT) [90]. Data from a functional assay established that knockdown of CD44 significantly reduces migration and invasion via repression of the ERK/Snail pathway, subsequently leading to decreased metastasis upon tail vein injection of KD CD44 of HCC cells into a metastatic mouse model [41]. CD44 is often combined with other CSC markers to enrich hepatic CSC populations, including CD133 and EpCAM. Upon knockout of CD44 in HCC, both CD133 and EpCAM are significantly downregulated [88]. Additionally, overexpression of the CD133^+^ CD44^+^ population in HCC cells promotes stem/progenitor cell properties, including extensive proliferation, self-renewal, differentiation, and increased resistance to chemotherapeutic agents, via upregulation of ABC transporters (ABCB1, ABCC1, and ABCG2) [21]. CD44 influences differentiated cells that experience carcinogenic insult to become proliferative cancer progenitors via activating AKT signaling and promoting Mdm2 translocation into the nucleus, with subsequent termination of the p53 response. In other words, HCC cells undergo DNA damage to escape p53-induced death and senescence and respond to proliferative signals that promote mutation and transmission to daughter cells that go on to become HCC progenitors [91]. Interestingly, an earlier study showed no correlation of CD44 with clinicopathological parameters in tumor tissues, with no differences in disease-free survival rates relative to the control group. However, in non-tumor tissues, the disease-free survival rate in the CD44-low expression group was significantly longer, compared to the CD44-high expression group. These initial findings clearly suggest that CD44 in non-tumor tissues is potentially associated with risk of tumor recurrence after hepatic resection for HCC [92].

### 2.6. CD13

CD13 (Aminopeptidase N) is a membranous glycoprotein and an extracellular peptidase, the major function of which is cleavage of single neutral amino acid from the N terminus of small peptides. However, CD13 plays a different kind of function and substrate depending on where it is expressed. For example, CD13 in the brain cleaves opioid peptide and enkephalins to regulate neuronal signaling. In the intestine, CD13 plays a critical role in cleaving peptides in the final digestion and resorption [93]. CD13 also acts as a candidate liver cancer stem cell marker. Microarray analyses have identified CD13 as an important candidate surface marker of the side population (SP) fraction. The CD13^+^ population was mainly detected in the G0/G1 phase and comprised dormant or slow-growing cancer cell populations correlated with chemoresistance through ABC transporter expression and recurrence. Additionally, CD13 is localized predominantly in the G0/G1 phase of the cell cycle, suggesting a role as a marker of the dormant or semi-stationary status of LCSCs [43]. Similar to CD44, CD13 could combine with other surface markers, including CD133 or CD90, and effectively initiate tumor formation, leading to increased HCC tumorigenesis [43]. CD13^+^CD133^+^ and CD13^+^CD90^+^ cells are reported to enhance not only HCC tumor initiation but also genotoxic chemoresistance to doxorubicin (DXR) and fluorouracil (5’Fu). Treatment with both DXR and 5’FU increased the CD13 population and subsequently reduced ROS-induced DNA damage after genotoxic stress along with inhibition of HCC apoptosis. [43]. TGF-β-induced EMT phenotypes and tumorigenicity in HCC are associated with elevated CD13 expression. Flow cytometry analyses have shown that cells positive for N-cadherin, an EMT marker, are localized in CD13-positive cell fractions, suggesting that TGF-β elicits both N-cadherin and CD13 expression. Additionally, CD13 could metabolize TGF-β/EMT-induced ROS and facilitate cell survival through evasion apoptosis, contributing to drug resistance [94]. Interestingly, CD13 can be induced not only by genotoxic chemotherapeutic agents but also by sorafenib in human HCC cells. Mechanistically, CD13^+^ CSCs are dependent on Tyr metabolism, rather than glucose, as an energy source. Simultaneously, Tyr metabolism produces nuclear acetyl-CoA. Stabilization and acetylation of Foxd3 are regulated by increasing CBP acetyltransferase activity through activation of the ERK1/2 signaling pathway [44]. Clinical data suggest that CD13 expression is correlated with that of EMT markers, such as E-cadherin and vimentin. Both disease-free and overall survival curves in high CD13 expression groups were significantly poorer than those of low CD13 expression groups. In view of the collective data on the association between CD13 and EMT markers, CD13 may serve as a prognostic marker of early recurrence after surgery [95]. 

### 2.7. CD90

CD90 (Thy-1) is a glycophosphatidylinositol (GPI)-anchored protein that is expressed in a variety of cell types, including T-cells activation [96], thymocytes, neurons outgrowth modulation [97], endothelial cells, the vesicular release of neurotransmitter at the synapse [98], astrocyte adhesion [99] and fibroblasts. In addition, CD90 is also involved in cytoskeleton organization, cell migration, and inflammation. CD90 is not only implicated in the tumorigenic and metastatic capacities of various HCC cell lines but also serves as a marker of LCSCs [45]. CD90^+^ cell populations have been isolated from both HCC specimens and blood samples of liver cancer patients and could initiate tumor nodule formation following intrahepatic injection into SCID/beige mice and subsequent secondary and tertiary transplantation into immunodeficient mice. These findings support the utility of CD90 as a surface marker for diagnosis of HCC malignancy [100]. Gene expression analysis of sorted cells has disclosed different features of EpCAM and CD90 populations. EpCAM possesses characteristics of epithelial cells while CD90 contains vascular endothelial cell features. Moreover, these markers are correlated with different clinicopathological parameters. Upon transplantation into xenografts, EpCAM^+^ cells promoted tumor growth in subcutaneous lesions and CD90^+^ cells induced high metastatic capacity in lung cells. Furthermore, CD90 influenced EpCAM^+^ cell motility through activation of the TGF-β pathway [61]. Cyclin D1 overexpression promoted stemness properties and enhanced chemoresistance through increasing CD90^+^ and EpCAM^+^ populations. Mechanistically, cyclin D1 interacted with and activated Smad2/3 and Smad4 signaling pathways to regulate LCSC self-renewal [48]. Circulating tumor stem cell (CTSC) populations within circulating tumor cells (CTC) play critical roles in the formation of distal metastatic tumors [101]. CD90^+^CXCR4^+^ present a better surface marker of CTSCs than CD133^+^CD90^+^, promoting the formation of tumor spheres in vitro, tumor development in primary and subsequent secondary and tertiary transplantation experiments, and distal metastatic tumors following subcutaneous transplantation [62]. Analysis of clinical HCC patients confirmed that CD90 is an important surface marker that is positively correlated with a number of clinicopathological parameters, including histopathology grade and tumor size. Additionally, CD90 expression is significantly associated with early recurrence and short-term survival of HCC patients [102,103]. Experiments on human tissue samples and JHH-6 HCC cell lines indicate that CD90 is significantly overexpressed in tumors and positively associated with growth factors, including hepatocyte growth factor (HGF), fibroblast associated protein (FAP), and alpha smooth muscle actin 2 (ACTA2). Moreover, CD90 induces chemoresistance to doxorubicin through reducing ABCB1 and increasing the ABC transporter proteins ABCG2 and Oct4, [46,47]. CD90 expression has been detected in not only LCSCs but also tumor-associated fibroblasts (CAFs). CD90 is positively correlated with clinicopathologic characteristics, including pathological grade, satellite lesions, PVTT, and recurrence, suggesting a role as a predictor of poor prognosis in CAFs, and is consistently determined as a reliable biomarker for prognosis of HCC patients subjected to hepatic resection [104].

### 2.8. CD24

CD24 is a mucin-like cell surface glycoprotein expressed in stem/progenitor cells and a variety of human malignancies, including hepatocellular carcinoma, breast cancer, renal cell carcinoma, colon cancer, B-cell lymphoma, differentiating neuroblasts, small cell and non-small cell lung carcinoma, and nasopharyngeal carcinoma. In addition, CD24 plays an important role in regulating neural development by contributing to the downstream signaling network by glycosyl- phosphatidylinositol (GPI) link to the cell surface [105]. CD24 expression has been reported in undifferentiated bipotential mouse embryonic liver stem and 3,5-diethoxycarbonyl-1,4-dihydrocollidine-treated oval cells and can help differentiate progenitor/stem cells from normal adult liver. Moreover, CD24 enhances expression of CK19, epithelial cell adhesion molecule, Sox 9, and FN14, which facilitate differentiation into hepatocytes [106]. These earlier studies suggest that CD24 is involved in metastasis, differentiation, self-renewal, and chemoresistance of HCC cells. Consistent with these findings, CD24 expression was shown to be positively correlated with metastasis in the HCC cell lines MHCC97H and HCCLM3, which display enhanced proliferation, migration, and invasive properties through increasing the CD24 population. Moreover, CD24 expression is associated with tumor number, tumor size, vascular invasion, encapsulation, differentiation, satellite lesions, and poor TNM stage in overall and relapse-free survival. CD24 is positively associated with PCNA and β-catenin, correlated with clinicopathologic features, including low AFP levels, and confers a greater propensity for multiple tumors and liver cirrhosis. Simultaneously, CD24 expression influences type II and III tumor recurrence. Taken together, the data clearly support the value of CD24 as a prognostic factor for HCC after surgery [49]. CD24 is proposed to play an important role in tumor-initiating cells (T-ICs) from chemoresistant (cisplatin) hepatocellular carcinoma (HCC) xenograft tumors orthotopically transplanted into SCID mice, promoting tumor-forming and self-renewal abilities through phosphorylation of STAT3 and induction of NANOG expression. mRNA microarray analyses further showed that the CD24 level is increased in chemoresistant groups despite no changes in other liver T-IC markers, including CD133, CD90, and EpCAM [50]. In an earlier investigation, CD24^+^ cells were isolated from two patient samples, followed by injection of 4000 CD24^+^ cells into NOD/SCID mice, which led to higher tumor development capability than CD24^−^ cells, supporting the tumorigenic property of CD24 [50]. In addition, CD24 induced resistance to both cisplatin and sorafenib. CD24 is more highly expressed in sorafenib-resistant relative to untreated wild-type cells, conferring greater resistance through activating autophagy and inhibiting the AKT/mTOR signaling pathway [51]. CD24^+^ LCSCs display enhanced levels of inducible nitric oxide synthase (iNOS), which triggers the Notch1 signaling pathway in a manner dependent on cGMP/PKG-mediated activation of TACE and upregulation of iRhom-2, further promoting self-renewal and tumor growth properties [52].

### 2.9. OV-6

OV-6 is a monoclonal antibody raised against cells isolated from carcinogen-treated BALB/c mouse liver [53,107]. The OV-6 population obtained from HCC cell lines and primary HCC tissues acts as a hepatic progenitor marker [108] and promotes self-renewal with endogenous activation of Wnt/β-catenin signaling. In other words, OV-6 expression is enriched in tumor cells via Wnt activation while inhibition of β-catenin signaling decreases the OV-6^+^ population in HCC. Moreover, OV6^+^ HCC cells display enhanced resistance to chemotherapeutic drugs, such as cisplatin [53]. OV6 promotes HCC tumors and leads to increased self-renewal capacity, tumorigenicity, and invasive and metastatic properties through positive correlation with C-X-C chemokine receptor type 4 (CXCR4) and its specific ligand, CXCL12 (SDF-1) [109]. The same research group proposed that HBx influences HCC self-renewal, stem cell-associated gene expression, tumorigenicity, and chemoresistance through stem-like properties of the OV6^+^ population. In terms of the underlying mechanism, HBx directly interacts with MDM2 to form a protein complex and subsequently inhibits MDM2 ubiquitin-directed self-degradation. Translocation of the HBx-MDM2 complex into the nucleus enhances transcriptional activity and expression of CXC4 and CXC12 and subsequent activation of the Wnt/β-catenin signaling pathway promotes the stem-like properties of OV6^+^ CSCs in liver [54]. 

### 2.10. Side Population Cells

Side population (SP) cells are originally shown to be enriched in stem cell compartments of various tissues and organs [110,111]. Subsequently, SP cells were detected in hepatocellular carcinoma cell lines, including Huh7 and PLC/PRF/5, using Hoechst 33342 dye staining in flow cytometry. The results suggested that SP cells may effectively serve as markers to distinguish between LCSCs and HCC cells and initiate tumorigenesis by upregulating stemness genes along with tumor formation in xenograft transplant experiments [112]. Microarray analysis of HCC cell lines demonstrated that other genes are additionally influenced in SP, compared with non-SP cells, such as GATA6 which is associated with embryonic development and hepatocytic differentiation and some ABC transporter genes, including ABCG2, ABCB1, and CEACAM6, linked to chemoresistance to doxorubicin, 5-fluorouracil, and gemcitabine, in addition to epithelial and mesenchymal markers. SP cells are therefore potentially involved in differentiation, chemoresistance, and metastasis of HCC [24,55]. Among these, ABCG2 in HCC cell lines is particularly significant. Flow cytometry experiments to isolate the ABCG2+ subpopulations of HCC cell lines sub-cultured for 4 weeks revealed the presence of both ABCG2^+^ and ABCG2^−^ progenies, supporting critical roles of ABCG2 in maintenance of the cancer cell hierarchy of HCC [113]. The researchers further showed intrinsic expression of ABCG2 in HCC tissues and cell lines. Furthermore, ABCG2 is reported to significantly influence drug efflux-related chemotherapy resistance in SP cells by altering subcellular localization through activation of the Akt signaling pathway [23].

### 2.11. CD47

CD47 is a transmembrane member of immunoglobulin, also as an integrin-associated protein (IAP), which is the expression in platelets, and binds to the signal-regulatory-protein-α (SIRPα) followed by inhibition of phagocytosis [57,114]. CD47 has been indicated to not only play an important role in immune evasion but to also regulate tumor apoptosis, angiogenesis, metastasis, tumor-initiating ability, chemoresistance, and proliferation in many cancers, including HCC. CD47 is overexpression in the HCC tumor cells and positively correlates with CD68^+^ (which is a macrophages marker) expression. Further, the results suggested that IL-6 derives from tumor-infiltrating macrophages that can induce CD47 expression in HCC by activation of the STAT3 signaling pathway. In addition, the same group also suggested that blocking of CD47 and combination with chemotherapeutic drugs enhance macrophage-mediated phagocytosis, which means lower expression of CD47 benefits the treatment with adjuvant transcatheter arterial chemoembolization (TACE) of HCC patients [58]. In addition, CD47 is one of the TICs markers, which shows the highest overexpression compared to other TICs markers in the sorafenib resistance cells. Both the clinical HCC sample and sorafenib resistance suggested positive correlations between NF-κB and CD47. CD47 is regulated by NF-κB, which can specifically bind to the CD47 promoter, and then up-regulation CD47 transcription activity follows [56]. CD47 in HCC is one of the LCSCs that has been developed of antibodies and widely studied. CD47 is blocked by an anti-CD47 monoclonal antibody, causing not only inhibition of self-renewal, tumorigenicity, migration, and invasion abilities but also synergizes in combination with chemotherapeutic drugs, including doxorubicin and cisplatin [115].

## 3. Interactions of LCSCs Influencing HCC and Therapeutic Strategies 

### 3.1. The Wnt/β-Catenin Pathway

Wnt/β-catenin signaling is highly and evolutionarily conserved in normal cells and participates in tissue homeostasis [116]. The canonical Wnt/β-catenin pathway is critical for HCC progression and tumorigenesis. β-catenin is a functional protein that plays a dual role in cell–cell adhesion and gene transcription and serves as an intracellular signal transducer in the Wnt signaling pathway [117,118,119,120]. Moreover, β-catenin is one of the subunits of cadherin that binds E-cadherin. However, mutation and overexpression of β-catenin promotes tumor progression in many cancer types, including hepatocellular carcinoma [121]. Under normal conditions, β-catenin in the cytosol is marked for ubiquitin-mediated proteolysis by specific phosphorylation of serine residues through an enzymatic complex including adenomatous polyposis coli (APC), Axin, and the kinases glycogen synthase kinase-3β (GSK-3β) and casein kinase I [122]. Wnt protein directly interacts with cell surface Frizzled receptors and LRP5/6 co-receptors upon activation of the Wnt/β-catenin pathway. Immediately after this step, Dishevelled protein is activated and released, leading to formation of the destructive enzymatic APC/Axin/GSK-3β complex and inhibition of GSK-3β. Further accumulation and stabilization leads to β-catenin translocation from the cytoplasm to nucleus and subsequent binding to TCF/LEF proteins, which activates downstream target genes, including MMP3, MMP7, ADAM10, Twist, Slug, Tiam1, c-Myc, cyclin D1, and fibronectin (Figure 2A) [123,124]. Frequent hyperactivation of the Wnt/β-catenin pathway in HCC patients leads to accumulation of β-catenin in tissue and nucleus, which serves as a hallmark of Wnt signaling. Activation of Wnt/β-catenin signaling has been shown to promote self-renewal, differentiation, and invasiveness in LCSCs [30,63,70,71]. Inhibition of Wnt/β-catenin signaling via small-molecule inhibitors reduces expression of Wnt and β-catenin proteins in LCSCs and further suppresses HCC stemness. One identified small-molecule inhibitor of Wnt/β-catenin signaling is CWP232228, an antagonist that competes with β-catenin for binding to TCF in the nucleus and suppresses transcriptional activity of downstream genes. This compound inhibits HCC self-renewal and tumor initiation through suppression of gene (Oct4, KLF4, Nanog, and SOX2) and surface marker (CD133^+^/ALDH1^+^) expression of LCSCs (Figure 2A.) [125]. FH535 is another antagonist that inhibits the Wnt/β-catenin signaling pathway and peroxisome proliferator-activator receptor (PPARγ and PPARδ) signaling [126]. This compound suppresses proliferation and motility of HCC cells through significant downregulation of cyclin D1 and MMP9 mRNA [127]. Additionally, FH535 is a potent therapeutic inhibitor that, upon combination with Sorafenib, exerts synergistic inhibition of proliferation and induction of apoptosis via enhancing cleaved poly (ADP-ribose) polymerase (PARP), inhibiting cyclin D1, Bcl-2, survivin, and c-MYC levels and reducing both mitochondrial respiration and glycolytic rates to disrupt the bioenergetics of HCC/LCSC cells [128,129]. WM130, a derivative of matrine, a Sophora alkaloid, has been shown to exert better pharmacological activities and anticancer effects against HCC than its parent compound. Results to date suggest that WM130 suppresses proliferation and self-renewal capability in both HCC and doxorubicin-resistant hepatoma cells by decreasing phosphorylation of GSK3β and subsequent degradation of β-catenin through downregulation of the CSC biomarker, EpCAM, and other stemness-related genes. Additionally, combined treatment with WM130 and doxorubicin synergistically inhibits tumor growth [130]. WM130 not only influences proliferation but also inhibits invasion and migration and induces apoptosis of HCC cells via suppression of EGFR/ERK/MMP-2 and PTEN/AKT signaling pathways [131]. Another inhibitor derived from matrine, (6aS, 10S, 11aR, 11bR, 11cS)-10- methylamino-dodecahydro- 3a,7a-diazabenzo (de) (MASM), inhibits the PI3K/AKT/mTOR and AKT/GSK3β/β-catenin pathways, with subsequent reduction of Bcl-2 and cyclin D1 expression along with increased PARP cleavage and p27 expression, and markedly reduces the EpCAM^+^/CD133^+^ cell population. These combined effects achieve inhibition of cell proliferation, induction of apoptosis, and cell cycle arrest, in addition to suppression of xenograft tumor growth for HCC [132]. Two anti-FZD antibodies are currently in clinical trials. One is OMP-18R5, which can interact with 5 out of 10 FZD receptors and competitively block Wnt3A interactions with FZD receptors. This compound has been shown to suppress tumor growth in a mouse model and exerts a greater inhibitory effect on tumor growth and delay in tumor recurrence in combination with a chemotherapeutic agent. OMP-18R5 has recently been evaluated in phase 1 clinical trials in patients with lung, breast, and pancreatic cancer [133]. The other preclinical anti-FZD antibody is OMP-54F28, a truncated FZD8 receptor fused to the IgG1 Fc region that competes with Fzd8 receptor for ligand binding, leading to suppression of the Wnt/β-catenin signaling pathway, inhibition of solid tumor growth, and decrease in CSC frequency, either alone or in combination with other chemotherapy drugs, such as gemcitabine [134]. Evaluation of the safety and pharmacokinetics/pharmacodynamics of OMP-54F28 combined with sorafenib in HCC patients has been completed in phase 1b clinical trials (ClinicalTrials.gov identifier: NCT02069145) [133]. The results suggested that the success of providing completed the safety and pharmacokinetics/pharmacodynamics of FZD-targeted therapy in HCC patients in phase 1 trials as well as suggested potential combinations of FZD-targeted and FDA-approved targeted therapy in HCC patients.

### 3.2. Notch Signaling 

The Notch pathway is a highly conserved cell signaling mechanism in multicellular organisms that regulates proliferation, maintenance of stem cells, differentiation, neurogenesis, embryonic development, maintenance of adult tissue homeostasis, and angiogenesis. The Notch receptor is a single-pass transmembrane receptor protein composed of functional extracellular (NECD), transmembrane (TM), and intracellular (NICD) domains. The protein exists as four isoforms, designated NOTCH1, NOTCH2, NOTCH3, and NOTCH4 [135,136], which interact with transmembrane ligands, such as Delta and Serrate, on neighboring cells. Ligand binding to NOTCH triggers cleavage and release of the Notch intracellular domain (NICD) and promotes translocation of the transcription factor complex (CBF1/RBPjk/Su(H)/Lag1 (CSL)) from the cytoplasm to nucleus, in turn, stimulating downstream target gene transcription (Figure 2B) [137]. Tumor progression, self-renewal, and CSC differentiation are influenced by Notch signaling, which is activated in most types of cancer. Accumulating research has focused on eliminating CSCs via targeting Notch signaling as a therapeutic strategy for cancer [137]. Several LCSC-related proteins, including CD90, Notch1, Nanog, and Sox2, which are overexpressed in parenchymal hepatic cells, have been identified via IHC. However, only CD90 was exclusively detected in HCC patient samples, supporting a correlation of this biomarker with poor prognosis. Isolated CD90^+^ populations from HCC cell lines were subsequently shown to exhibit increased tumorigenicity, chemoresistance, tumor invasion, and metastasis through Notch signaling activation. Moreover, the Notch signaling pathway promoted self-renewal, invasion, and migration of CD90^−^ cells. The collective data suggest that CD90 is an effective biomarker for LCSCs and specifically upregulates stem-associated genes Nanog, Oct4, and Sox2 through activated Notch signaling [138]. C8orf4 deletion promotes nuclear translocation of N2ICD and subsequently triggers Notch2 signaling, which enhances CD133^+^/CD13^+^ expression and sustains LCSC stemness [139]. Notch3 is overexpressed in HCC patients and positively correlated with clinicopathological parameters, including alpha-fetoprotein (AFP) levels, poor prognosis (shorter survival time), and cisplatin resistance. Moreover, high Notch3 is correlated with lower expression of β-catenin but higher aldehyde dehydrogenase (ALDH) activity, supporting the theory that Notch3 regulates tumor stemness through inactivation of Wnt/ β-catenin [140]. Notch and Wnt/β-catenin signaling pathways are intercalated and play critical roles in stemness characteristics and metastasis of LCSCs. Notch1 is upregulated and activates Notch1 intracellular domain (NICD) expression in HCC through Wnt/β-catenin signaling. Treatment with the Wnt/β-catenin-specific tankyrase1/2 inhibitor (XAV939) and γ-secretase inhibitor (DAPT) that block the Wnt/β-catenin and Notch signaling pathways, respectively, leading to suppression of tumor growth [141]. PF-03084014 is a γ-secretase inhibitor that blocks self-renewal and proliferation of cancer stem cells. The compound has been shown to reduce both hepatocellular carcinoma tumors growth and metastasis in sphere-derived orthotopic tumor model and one patient-derived xenograft (PDX) model. Additionally, low-dose PF-03084014 induces sphere differentiation of hepatocellular carcinoma and further reduces chemoresistance, supporting its value as a novel antitumor and antimetastatic therapeutic agent for HCC [142].

### 3.3. Hedgehog Signaling Pathway

The hedgehog signaling (Hh) pathway facilitates normal development of mammalian embryos and regulates cell proliferation, survival, and differentiation. Hh is limited to stem cell subsets that undergo rapid turnover and modulate tissue repair in adult tissue, such as nervous system, skin, and intestines [143,144,145,146]. The canonical Hh pathway is activated by three ligands, specifically, Desert hedgehog (DHH), Indian hedgehog (IHH), and Sonic hedgehog (SHH), which interact with the 12-pass transmembrane protein receptors Patched 1 (PTCH1) and Patched 2 (PTCH2). In the absence of ligand, Smoothened (SMO), a 7-pass transmembrane G-protein-coupled signal transduction molecule, is localized in vesicles. Ligand binding to PTCH1 or PTCH2 promotes SMO localization to primary cilium on the cell membrane. This step initiates translocation of glioma-associated oncogene homolog (GLI), a transcription factor, to the nucleus and subsequent activation of the downstream genes GLI1, PTCH1, cyclin D, VEGF, and c-myc (Figure 2C). However, upon overexpression of Hh ligands, loss of function of the receptor or abnormal transcription factor expression leads to dysregulation of the Hh pathway and further initiation and progression of multiple cancer types, including breast cancer, prostate cancer, hepatocellular carcinoma, pancreatic cancer, and brain cancers [147]. In clinical HCC cases with both HBV infection and HBx transgenic mice (HBxTg), correlations with Hh marker upregulation have been reported. Therefore, blocking the Hh signaling pathway may inhibit HBx-induced migration, anchorage-independent growth, and HBxTg tumor development [148]. A number of studies have shown that Hh markers are overexpressed in HCC patients and positively correlated with clinicopathological parameters (tumor differentiation, encapsulation, vascular invasion, early recurrence, and intrahepatic metastasis) along with significantly poorer overall and disease-free survival [149,150]. Dysregulation of Hh signaling contributes to maintenance of stemness in CSCs and influences tumor growth, metastasis, drug resistance, and differentiation [151]. In a highly tumorigenic CD133^+^ population of HCC, Smoothened (SMO) is abundantly expressed in association with the Hh signaling pathway and influences liver cancer stemness maintenance [35]. Current development of therapeutic agents that target Hh signaling has primarily focused on natural and synthetic antagonists of SMO and GLI1, which have undergone clinical trials with varying degrees of success [152]. Cyclopamine, an isolate of *V. californicum*, was initially shown to inhibit the Hh pathway by direct binding to SMO [153]. Cyclopamine inhibits tumor proliferation and growth in multiple cancer types, including HCC. The compound is reported to suppress Hep3B hepatocarcinogenicity by inhibiting SMO and blocking overactivation of Hh signaling by 50%, along with decreasing c-myc transcription activity [154]. However, cyclopamine-induced reduction of tumor growth is associated with several adverse side effects, including weight loss, dehydration, and death, in a mouse model [155,156]. GDC-0449, a second-generation cyclopamine, also directly binds SMO to prevent GLI activation with subsequent inhibition of the Hh pathway [157]. The compound was approved by the FDA as the first Hh signaling inhibitor for treatment of primary or recurrent cancers. GDC-0449 induces a significant decrease in liver myofibroblasts, progenitors, and liver fibrosis through inhibitory effects on Hh signaling, leading to suppression of HCC metastasis with no adverse effects on mortality [158]. These results support the utility of GDC-0449 as a safe clinical therapeutic agent for cancer. Moreover, activation of Hh signaling enhances the liver macrophage-mediated proinflammatory response and critical progression of nonalcoholic fatty liver disease (NAFLD). The overall findings clearly indicate that GDC-0449 and LED225 effectively reduce macrophage activation and proinflammatory cytokine expression (TNF-α, IL-1β, MCP1, and IL-6) as well as further NAFLD progression through direct binding to SMO and blocking of the Hh signaling pathway, supporting their potential utility as effective therapy for NAFLD-induced liver cancer [159]. LDE-225 is a second-line treatment agent for cancer approved by the FDA that exerts its effects by inhibiting the Hh signaling pathway. Data from Phase I/II clinical trials support the efficacy of LDE-225 as both monotherapy and combination therapy for numerous solid or hematological malignancies, including hepatocellular carcinoma (HCC) and Child-Pugh A cirrhosis (CPA). Phase I trial results indicate that the maximum safe dose of LDE-225 can be used to effectively treat advanced or metastatic HCC and CPA (ClinicalTrials.gov Identifier: NCT02151864). GANT-61 is a Gli transcription inhibitor that modulates genes involved in cell proliferation, self-renewal, and metastasis, resulting in suppression of pancreatic CSC characteristics and tumor growth. The compound is reported to show efficacy as a chemotherapeutic agent for human pancreatic cancer by regulatory activity on apoptosis markers [160]. The above studies have reported that the Hh signaling pathway is activated in HCC and effectively targeted by related small-molecule inhibitors for therapeutic purposes. Additionally, the group of Wang [161] showed that GANT61 induces autophagy that contributes to apoptosis and, further, suppression of HCC tumor growth. Clearly, autophagy plays an important role in determining the response to Hh-targeted therapies for HCC.

### 3.4. TGF-β Signaling Pathway

TGF-β family ligands not only play critical roles in cell proliferation, growth, homeostasis, and differentiation but also influence self-renewal of many stem cell types. TGF-β proteins initiate intracellular signaling by binding to the surface protein complexes of transmembrane kinases, which can distinguish between TβRI and TβRII. Interactions with ligand stabilize both TβRI and TβRII. Subsequently, TβRII phosphorylates the GS domain of TβRI, leading to its activation. Subsequent recruitment of the intracellular SMAD protein and translocation to the nucleus triggers downstream gene transcription (Figure 2D) [162]. TGF-β plays a dual role in liver cancer development, acting as a tumor suppressor at the early stages while promoting metastasis in the late stages of cancer progression [163,164]. Tumor-associated macrophages (TAM), a critical component of immune cells in the tumor microenvironment are positively correlated with the EpCAM^+^ population and enhance cancer stem cell-like properties via upregulation of CSC transcription factors Bmi1 and Klf4. However, blockage of TGF-β expression using anti-TGF-β antibodies significantly inhibits stem cell-like behavior via downregulation of Bmi1 and Klf4 transcription [165]. In addition, TGF-β displays the capability to induce CD133 and further promotes tumor initiation upon subcutaneous inoculation into nude mice [166]. LY2157299, also known as galunisertib, specifically inhibits TGF-β receptor (TβR)-I activation and exerts anti-proliferative and anti-invasive effects against HCC. Treatment of IHC with LY2157299 reduced expression of the proliferation marker, Ki67, and induced that of caspase-3, an apoptosis marker. In patient samples, LY2157299 could effectively inhibit proliferation and induce apoptosis upon administration as both monotherapy and in combination with Sorafenib. LY2157299 has been shown to exert synergistic therapeutic effects with Sorafenib [167]. However, the utility of LY2157299 in LCSCs remains to be established. Furthermore, a number of small-molecule inhibitors of TGF-β signaling have been developed with a specific focus on Smad protein, which effects LCSCs expression in HCC. GP73 is reported to enhance Smad2/3 phosphorylation via activation of TGF-β1 and promote EMT via upregulation of EMT marker expression while SB431542 specifically inhibits Smad2/3 phosphorylation and reverses EMT in HCC [168]. Cyclin D is dependent on activation of Smad2/3 and Smad4, leading to enhancement of single sphere formation, CD90^+^ and EpCAM^+^ populations, stemness gene expression, and chemoresistance (Figure 2D). Smad inhibitors impaired cyclin D-induced self-renewal, enhanced sensitivity to chemotherapy, and suppressed tumor growth in a cyclin D sphere-derived xenograft model [48].

### 3.5. Targeting of LCSC Surface Markers

Several therapeutic surface markers have been identified to date. CD133 is a surface marker isolated from tumor-initiating cells (TICs), which is significantly correlated with poor prognosis of HCC patients. Two types of oncolytic measles viruses (MV), MV-141.7 and MV-AC133, have been described that target CD133 and selectively destroy CD133^+^ tumor cells [169]. Moreover, enhanced oncolytic activities and prolonged survival of CD133-targeted viruses in clinical trials, compared with parental MV-Nse, support their close association with oncolytic agents. VB4-845, a recombinant fusion protein, is an anti-EpCAM single-chain antibody. In eight HCC cell lines, VB4-845 strongly suppressed sphere formation ability, compared to 5-FU. In addition, in high EpCAM^+^ expressing cell populations, VB4-845 suppressed cells expressing CD133 and CD13 while treatment with 5’FU induced CD133- and CD13-positive populations [170]. Evaluation of VB4-845 in clinical trials for various cancer types, including urothelial carcinoma, bladder cancer, squamous cell cancer, and head-and-neck cancer, support its therapeutic efficacy for HCC [171]. Isolated CD90^+^CD44^+^ cells from HCC displayed better tumorigenic capacity than CD90^+^CD44^−^ cells. Thus, a CD44-antibody could effectively target CD90^+^CD44^+^ cell population and induce apoptosis in HCC [45]. In another study, CD44 antibody-mediated liposomal nanoparticles in HCC promoted apoptosis and reduced tumor growth through specifically targeting CD44 [172]. A further potential therapeutic strategy is development of surface marker inhibitors, such as Ubenimex, which suppress CD13 and are widely used as an immunoenhancer for treating solid tumors and hematological neoplasms. Ubenimex is reported to reverse multidrug resistance (MDR) of HCC by suppressing Pim3, BCL-2, and BCL-XL expression, decreasing Bad phosphorylation and further enhancing tumor cell apoptosis [173].

### 3.6. Epigenetic Changes

Alterations in epigenetic control of gene expression play a substantial role in disease progression, including hepatocellular carcinoma (HCC) [174]. Epigenetic modification mechanisms include DNA methylation, histone modification, and chromatin remodeling. Notably, epigenetic changes can influence activation of certain genes but not the genetic code of DNA [175,176,177]. Recent studies have suggested that epigenetic changes are correlated with LCSC phenotypes. For instance, suppression of DNA methylation via treatment with the DNMT1 inhibitor, zebularine (ZEB), induced highly tumorigenic cells within the side population (SP) fraction, leading to increased tumorigenicity of HCC [178]. Treatment of high-density cells with ZEB reduced expression of surface markers (such as CD133, CD44, EpCAM) and cancer stem cell properties as well as tumorigenesis, concomitant with upregulation of genes related to apoptosis and differentiation. Taken together, the results suggest that DNA methylation acts as a key epigenetic regulatory factor of LCSCs (Figure 2E) [179]. In clinical HCC patients, SALL4 was positively correlated with EpCAM and significant decrease in overall survival [60]. SALL4 is highly expressed in HCC and associated with elevated serum alpha-fetoprotein levels, high frequency of hepatitis B virus infection, and poor prognosis. In addition, SALL4 is reported to induce spheroid formation and upregulation of LCSC markers, including KRT19, EPCAM, and CD44, in HCC. HDAC activity was induced in SALL4-positive cells via SALL4 interactions with nucleosome remodeling and deacetylase (NuRD) complexes. The HDAC inhibitor, SBHA, may present an effective option to inhibit SALL4-positive cell proliferation [180].

### 3.7. MicroRNAs and Long Non-Coding RNAs

Evidence from a systematic review suggests that high-resolution microarray and deep sequencing techniques have encouraged studies on non-coding RNAs (ncRNAs) including short non-coding RNAs (miRNAs and sRNA) (<200 nucleotides) and long non-coding RNAs (lncRNAs) (>200 nucleotides). In human plasma and tumor tissues of HCC, several miRNAs and lncRNAs that regulate LCSCs show promise as biomarkers for diagnosis and prognosis as well as response to therapy. Both miRNAs and lncRNAs can act as critical regulators of self-renewal, differentiation, and tumorigenicity of CSCs. For example, microRNA-150 upregulated in a CD133- subpopulation of HCC interacts with 3’UTR of c-myc mRNA, consequently inhibiting c-myc protein levels. Overexpression of microRNA-150 significantly reduced the CD133^+^ cell population and further inhibited cell growth and tumor sphere formation, inducing cell cycle arrest and apoptosis [181]. MicroRNA-449a is overexpressed in poorly differentiated HCC tissues, drug-resistant cell lines, and Nanog-positive liver cancer cells. Upregulation of microRNA-449a is proposed to increase stemness in HCC. Notably, knockdown of miR-449a via transcription factor 3 (TCF3) reduced Nanog transcription and influenced cellular stemness [182]. Oct4/miR-1246 in CD133^+^ HCC is correlated with a poor diagnosis which promotes cancer stemness through activation of the Wnt/β-catenin pathway through miR-1246 targeting to degradation of β-catenin complexes suppressing AXIN2 and GSK3β expression [183]. miR-125b is down-regulated in HCC by TGF-β1-induced and negatively correlated with LCSCs expression. Besides, in vitro and in vivo studies suggested that EMT and EMT-associated traits of HCC are inhibited by targeting SMAD2 and SMAD4 by ectopic expression of miR-125b [184]. miR-216a/217 prompted migration and metastatic ability and increased the stem-like cells population and tumor recurrence in HCC by activation of PI3K/Akt and TGF-β pathways by induction of targeting PTEN and SMAD7 expression [185]. Recent findings by our group suggest that high BC200 expression in the CD133^+^ population of HCC promotes tumor growth and sphere formation, supporting its requirement in self-renewal and maintenance of LCSCs [186]. LncSox4 is highly expressed in HCC tissues and liver tumor-initiating cells (TICs) and required for liver TIC self-renewal and tumor initiation. LncSox4 regulates TIC self-renewal by recruiting STAT3 that binds the Sox4 gene promoter and induces Sox4 transcription activity. Simultaneously, Sox4 is positively correlated with expression of the surface markers EpCAM and CD133 that promote TIC self-renewal [187]. Both lncRNAs and Wnt/β-catenin signaling pathways influence LCSCs of HCC. HCC tumors and LCSC-containing HCC cell lines (CD13^+^CD133^+^) highly express LncTCF7, a critical regulator of LCSC self-renewal and tumor propagation. LncTCF7 recruits SWI/SNF complex binding to the TCF7 promoter and triggers TCF7 transcription and further activation of Wnt signaling-induced tumorigenic activity in liver cancer stem cells [188]. The long non-coding RNA, MEG3, a tumor suppressor in HCC, is reported to promote β-catenin degradation through increasing PTEN, leading to inhibition of hepatocarcinogenesis [189]. Lnc-β-Catm is promoted β-catenin stabilization by methylation of β-catenin by methyltransferase EZH2, thereby regulation of self-renewal of LCSCs [190]. LncDANCR in the stem-like cells of HCC is overexpression and further increases stemness features and promotes tumorigenesis of HCC via interacting with CTNNB1 mRNA and derepression of CTNNB1 [191]. Overexpression of LncDILC has dramatically decreased LCSCs expression by down-regulation of IL-6 transcription and abrogation of STAT3 activation then inhibition of IL-6/STAT3 signaling [192]. LncHDAC2 is highly expressed in CD13^+^CD133^+^ cells, which promotes LCSC self-renewal and tumor progression by recruiting NuRD complex binding to the promoter of PTCH1, leading to inhibition of transcriptional activity and subsequent activation of Hh signaling [193]. The collective findings support the potential utility of ncRNAs as therapeutic targets against LCSCs in liver cancer.

### 3.8. The LCSC Microenvironment

Accumulating studies have shown that in addition to genetic, epigenetic, and signaling pathways, the microenvironment (niche) is an important influencer of stem cell behavior and differentiation. For instance, embryonic stem cells can significantly suppress tumorigenicity of aggressive cancer cells induced by the microenvironment [194]. Similarly, the tumor microenvironment influences cancer stem cell characteristics, leading to malignant phenotypes. For example, adding conditioned medium of mouse Lewis lung carcinoma to Nanog miPS cells leads to acquisition of CSC features and subsequent formation of spheroids in suspension culture, along with induction of high tumorigenicity and angiogenesis ability in Balb/c nude mice [195]. Single-cell CSC clones from human liver cancer microvascular endothelial cells were isolated and further treated with different tumor cell-derived conditioned culture media mimicking the tumor microenvironment. The results showed differentiation into the corresponding tumor cells and expression of specific tumor cell markers [196]. The major component of tumor stromal cells is cancer-associated fibroblasts (CAFs), which positively express CD90 and CD44. Upon co-culture with the human HCC cell lines Huh7 and JHH-6, CAFs enhanced mRNA expression of TGFB1 and FAP, compared to non-tumoral fibroblasts (NTF) (Figure 2G). These findings suggest that CAFs and HCC interact with each other to function in the maintenance and progression of liver disease [197]. The α-SMA(+)CAF population is regulated with T-ICs via activation of FRA1, dependent on ERK signaling activation by HGF. Activated FRA1 binds to downstream genes, stimulating HEY1 promoter transcription activity, and further enhances HCC self-renewal and tumorigenesis. These findings suggest that T-ICs of HCC are regulated extrinsically within the tumor microenvironment [198]. Isolation of the CD44^+^ population from human HCC cells and incubation with TAM induced expansion of this cell population and tumor sphere formation. In addition, TAMs and co-culture with HCC cell lines promoted expression of the cytokine IL-6 and expansion of CD44^+^ cells. Knockdown of STAT3 or treatment with tocilizumab blocked IL-6 signaling and further inhibited TAM-stimulated activity of CD44^+^ cells (Figure 2G) [87]. In a hypoxic tumor microenvironment, rapid tumor growth further influences the proportion of LCSCs in HCC [199]. Artemis (ARTN), a hypoxia-responsive factor, is a necessary regulator promoting hypoxia-induced LCSC expansion of the CD133^+^ population in HCC, leading to enhanced tumor sphere formation and tumorigenesis [200]. Disulfiram (DS) is an anti-cancer drug that specifically inhibits the scavenger activity of reactive oxygen species (ROS) but has a very short half-life in the bloodstream. To solve this issue, a poly lactic-co-glycolic acid (PLGA)-encapsulated DS (DS-PLGA) was developed, which could inhibit the liver cancer stem cell population with a synergistic cell killing effect in combination with 5-FU or sorafenib. These findings suggest that the anti-HCC efficacy of DS is mediated through inhibition of LCSCs [201].

## 4. Conclusions

LCSCs can be identified based on a series of surface markers, including EpCAM, CD133, CD44, CD13, CD90, CD24, OV-6, CD47, and isolated SP cells. Functional studies suggest that LCSCs influence tumorigenesis, metastasis, and therapeutic resistance, as well as recurrence of HCC, and identification of this cell population should assist with diagnosis and prognosis predictions, patient stratification for administration of individualized therapy, and development of novel LCSC-targeted therapeutic strategies for HCC. LCSCs regulate tumor progression and therapeutic resistance via several mechanisms, including gene mutations, distribution of epigenetics, dysregulation of signaling, and alterations in the microenvironment. Small-molecule inhibitors against dysregulation of signaling pathways, such as Wnt/β-catenin, Notch, Hh, and TGF-β, may effectively suppress LCSC-mediated tumorigenesis, metastasis, and self-renewal. OMP-18R5 and OMP-54F28 (inhibitors of the Wnt/β-catenin signaling pathway), LED225 (a Hh inhibitor), and LY2157299 (a TGF-β inhibitor) are progressing to clinical trials alone or in combination with other chemotherapeutic agents targeting LCSCs of HCC [133]. Oncolytic measles viruses specifically targeting CD133 (MV-141.7 and MV-AC133) and anti-surface marker antibodies have additionally been developed for clinical trials, including anti-EpCAM (VB4-845) and anti-CD44, which block surface markers of LCSCs alone or in conjunction with other chemotherapeutic agents for HCC. Epigenetic regulation plays a critical role in every step of tumorigenesis and progression. Some LCSC features are influenced by epigenetic control, and therefore, inhibitors of epigenetic regulation mechanisms (such as zebularine and SBHA) can achieve significant suppression of tumor-inducing characteristics of LCSCs. However, epigenetic modulation is not effective for targeting specific genes, which could become non-specific alterations. A number of other factors influence LCSCs, such as the tumor microenvironment and non-coding RNAs (miRNAs and lncRNAs). Currently, combinations of chemotherapy agents and small-molecule inhibitors are being developed to reduce LCSC populations for effective treatment of HCC. Since LCSCs share a number of similar characteristics with stem cells, distinguishing between the molecular signaling pathways or mechanisms of the two cell subpopulations remains a considerable challenge, and thus treatment approaches that target LCSCs are limited. Further research focus on the biological differences between normal stem cells and LCSCs is warranted. Elucidation of the unique biological or phenotypic properties of LCSCs should facilitate the development of effective therapeutic agents without the need for normal stem cells.

## Figures and Tables

**Figure 1 cells-09-01331-f001:**
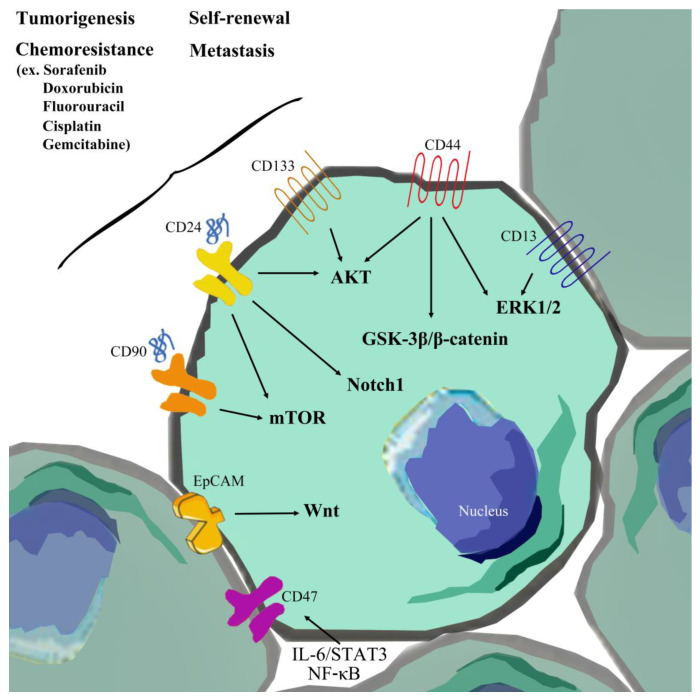
Activation of signaling pathways in liver cancer stem cells (LCSCs) influences hepatocellular carcinoma (HCC) development. Surface markers (including EpCAM, CD133, CD44, CD13, CD90, CD24, and CD47) influence the activation of signaling pathways, phenotypes, and resistance to clinical drugs in LCSCs

**Figure 2 cells-09-01331-f002:**
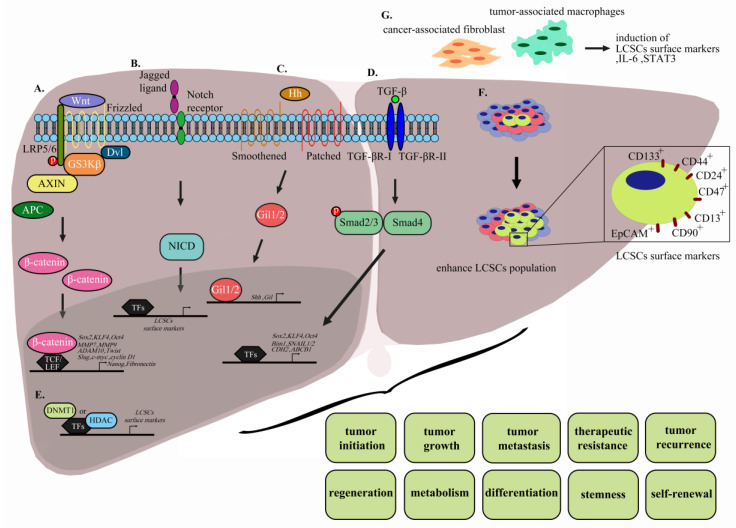
Interactions of LCSCs influencing HCC. (**A**) The Wnt protein directly interacts with cell surface receptor Frizzled and LRP5/6 co-receptors for activation of the Wnt/β-catenin pathway. Immediately after, Dishevelled protein is activated and released, leading to the generation of destructive enzymatic complex (APC/Axin/GSK-3β) and inhibition of GSK-3β. Following accumulation and stabilization, β-catenin translocates from the cytoplasm to the nucleus and subsequently binds TCF/LEF proteins to activate transcription of downstream target genes, including MMP3, MMP7, ADAM10, Twist, Slug, Tiam1, c-Myc, cyclin D1, and Fibronectin. (**B**) Ligand binding to NOTCH leads to cleavage and release of the Notch intracellular domain (NICD), promoting transcription factor complex (CBF1/RBPjk/Su(H)/Lag1 (CSL)) translocation from the cytoplasm to nucleus and activation of downstream target genes. (**C**) Desert hedgehog (DHH), Indian hedgehog (IHH), or Sonic hedgehog (SHH), the ligands binding PTCH1 or PTCH2, promote SMO localization to primary cilium on the cell membrane. Subsequently, glioma-associated oncogene homolog (GLI), a transcription factor, translocates to the nucleus and activates transcription of downstream genes. (**D**) Interactions with ligand stabilize TβRI and TβRII, following which TβRII phosphorylates the GS domain of TβRI, leading to further activation. Subsequent recruitment of intracellular SMAD proteins and translocation to the nucleus stimulate downstream gene transcription. (**E**) DNA methylation transferase (DNMT1) and histone deacetylases (HDAC) act as key epigenetic regulatory factors for downstream gene transcription. (**F**) Enhancement of surface markers of LCSC populations, including EpCAM, CD133, CD44, CD24, CD13, CD90, and CD47. (**G**) Cancer-associated fibroblast (CAFs) and tumor-associated macrophages (TAMs) promote LCSC surface marker populations (IL-6 and STAT3) within the microenvironment.

**Table 1 cells-09-01331-t001:** Surface markers influencing the signaling pathways, phenotypes, and resistance to clinical drugs in LCSCs.

LCSCs	Phenotypes of LCSCs (Source)	Signaling Involving LCSCs	Resistance to Clinical Drug	Ref.
EpCAM	cell–cell adhesion, metabolism, cell signaling, differentiation, metastasis, regeneration, organogenesis, tumorigenesis, chemoresistance and self-renewal (Hep3B, HepG2, Huh7, Huh1, and Dt81Hepa1-6 cells)	Activation of the Wnt signaling pathway	Sorafenib	[30,31,32,33]
CD133	tumorigenic, cell cycle progression, differentiation, chemoresistance, and self-renewal (Huh7, SMMC7721, PLC8024,PLC8024, HepG2, and HCCLM3 cells)	Activation of AKT/PKB,	Doxorubicin, Fluorouracil (5-FU) and Sorafenib	[27,34,35,36]
CD44	proliferation, survival, migration/invasion, and chemoresistance, and self-renewal (primary HCC, HepG2, Hep3B, Huh7, SUN-368, SUN-354, SMMC-7721, and MHCC97-H cells)	Activation of AKT/GSK-3β/β-catenin, and ERK/Snail pathways	Doxorubicin	[21,37,38,39,40,41,42]
CD13	chemoresistance, tumorigenesis and self-renewal (Huh7, PLC, and HepG2 cells)	Activation of ERK1/2 signaling pathway	Sorafenib, Doxorubicin, and Fluorouracil (5-FU)	[43,44]
CD90	tumorigenesis, metastasis, self-renewal and chemoresistance (MHCC97L, PLC, HepG2, Hep3B, primary HCC, and JHH-6 cells)	Activation of mTOR signaling pathway	Doxorubicin	[45,46,47,48]
CD24	metastasis, differentiation, self-renewal and chemoresistance (MHCC97H, HCCLM3, PLC/PRF/5, Huh7, and Hep3B cells)	Autophagy activation, activation of AKT/mTOR signaling pathway, and Notch1 signaling pathway	Cisplatin, Sorafenib	[49,50,51,52]
OV-6	self-renewal, tumorigenicity, and chemoresistance (SMMC7721, and HuH7 cells)	Activation of Wnt/β-catenin signaling	Cisplatin	[53,54]
Side population	differentiation, chemoresistance, and metastasis (Huh7, PLC/PRF/5, HCCLM3, MHCC97-H, MHCC97-L, and Hep3B cells)	Activation of AKT signaling pathway	Doxorubicin, Fluorouracil (5-FU), and Gemcitabine	[23,24,55]
CD47	self-renewal, tumor initiating, tumorigenicity, and chemoresistance (MHCC97L, PLC, and Huh7 cells)	Activation of IL-6/STAT3 signaling pathway, and NF-κB	Doxorubicin, Sorafenib	[56,57,58]
SALL4	proliferation, differentiation, and chemoresistance (Huh7, PLC/PRF/5, and patients of HCC)	Interaction with NuRD, regulation of PTEN, and PI3K/AKT signaling pathway	Fluorouracil (5-FU)	[59,60]
CD13^+^CD133^+^	tumor initiation, chemoresistance, and anti-apoptosis (Huh7 and PLC cells)	Reduction of ROS-induced DNA damage and inhibition of apoptosis	Doxorubicin, Fluorouracil (5-FU)	[43]
CD13^+^CD90^+^	tumor initiation, chemoresistance, and anti-apoptosis (Huh7 and PLC cells)	Reduction of ROS-induced DNA damage and inhibition of apoptosis	Doxorubicin, Fluorouracil (5-FU)	[43]
EpCAM^+^ CD90^+^	metastasis, tumorigenesis (patients of HCC and primary HCC)	activation of the TGF-β pathway		[61]
CD90^+^CXCR4^+^and CD133^+^CD90^+^	tumor development, tumor spheres, and metastasis (primary HCC)			[62]

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
