# Peer review of "Cancer Stem Cell Functions in Hepatocellular Carcinoma and Comprehensive Therapeutic Strategies"

_cells, 2020, doi:10.3390/cells9061331_

Round 1
Reviewer 1 Report
The review presented by Liu et al, is an extremly clear reppresentation of the surface markers for liver cancer stem cell targeted therapeutic strategies for HCC. Authors have clearlt and minutely reported the current literature. The figure sums up the descriptive text helping the readers to summarise the information.
Author Response
Response to Reviewer 1 Comments:
The review presented by Liu et al, is an extremly clear reppresentation of the surface markers for liver cancer stem cell targeted therapeutic strategies for HCC. Authors have clearlt and minutely reported the current literature. The figure sums up the descriptive text helping the readers to summarise the information.
Authors’ response: Thank you for the comment review of the manuscript “Cancer stem cell functions in hepatocellular carcinoma and comprehensive therapeutic strategies”.

Reviewer 2 Report
This is a good review of cancer stem cells in HCC development and comprehensive therapy. Though there are different reviews talking about cancer stem cells in HCC, the manuscript focused on the updates of various biomarkers and related signal pathways involving in HCC progress.
However, there still have some questions on the manuscript.
Firstly, could the author just spend a little space to update the cancer stem cell original research progress, since it is still in an argument.
Some abbreviations should not put the full name again since they were shown in the previous context.
Meanwhile, some biomarkers talked here are not specific biomarkers for cancer stem cells, so the author also should better mention a bitter on them.
About the part 2, about table 1, CD133+CD90+ or other double-positive cancer stem cells could be added, and how about the original source of the cell lines, which is very convenient for readers to choose in their study. Meanwhile, there are some other markers might be used in cancer stem cells, such as SALL4, CD47, et al.
To part 2, it would be better if the author could make a figure to link these stem cell markers with signal pathways in HCC development process.
About the figure 1, liver stem cells play roles in the tumor differentiation process, but the figure only emphasizes these stem markers involved in stemness and therapy resistance. Meanwhile, about the miRNA and non-coding RNA in LCSC regulating, both of them also showed some function on the signal pathway involved in FIG 1 A, B, C, D, E, the author should better add more related research reports.
Author Response
Response to Reviewer 2 Comments:
Point 1: Firstly, could the author just spend a little space to update the cancer stem cell original research progress, since is still in an argument.
Authors’ response: We followed the reviewer’s comments and added the new section to introduce the recent progress of cancer stem cell and the difficulty of CSC therapy. The new section of 2.1. Concept of cancer stem cells (CSCs) (p.2, track-change version), which discusses how to find and identify the origin of CSCs in cancer and provides the new concept. Further, we also introduce the relationship between CSCs’ biological function and cancer progression, and how do CSCs acquire the drug resistance and lead tumor relapse after therapy in the HCC cells and clinical treatments.
Point 2: Some abbreviations should not put the full name again since they were shown in the previous context.
Authors’ response: The abbreviations have been spelled out in the track-change version.
Point 3: Meanwhile, some biomarkers talked here are not specific biomarkers for cancer stem cells, so the author also should better mention a bitter on them.
Authors’ response: We followed the reviewer’s comments and added the introduction of function on some of the markers, which function is independent on CSCs (p.5-7, track-change version).
Point 4: About the part 2, about table 1, CD133+CD90+ or other double-positive cancer stem cells could be added, and how about the original source of the cell lines, which is very convenient for readers to choose in their study. Meanwhile, there are some other markers might be used in cancer stem cells, such as SALL4, CD47, et al.
Authors’ response: We have revised tables 1 in the manuscript and added double-positive cancer stem cells, the original source of the cell lines in every LCSCs, and CSCs marker of CD47 and SALL4 in tables 1 (p.8-10, track-change version). In addition, we also added a new section, which focuses on the discussion of the relationship between CD47 and HCC progression. The new section of 2.10. CD47 (p.8, track-change version), which is the introduction of CD47 biological function and focusing on the CD47 therapeutic strategy in the HCC.
Point 5:To part 2, it would be better if the author could make a figure to link these stem cell markers with signal pathways in HCC development process.
Authors’ response: We have added a new Figure 1 (p.10, track-change version), which summarizes the LCSCs and their activation of signaling pathways to influence the HCC development process.
Point 6: About the figure 1, liver stem cells play roles in the tumor differentiation process, but the figure only emphasizes these stem markers involved in stemness and therapy resistance. Meanwhile, about the miRNA and non-coding RNA in LCSC regulating, both of them also showed some function on the signal pathway involved in FIG 1 A, B, C, D, E, the author should better add more related research reports
Authors’ response: We have added the roles of miRNA (such as miR-1246, miR-125b, miR‐216a, and miR-217) (p.15-16, track-change version) and lncRNA (such as lnc-β-Catm, lncDANCR, lncDILC, and lncHDAC2) (p.16, track-change version) in LCSCs. Both the new adding miRNAs and lncRNAs are involved in the signaling pathway in figure 2A, B, C, D, E.
